# Epitope profiling using computational structural modelling demonstrated on coronavirus-binding antibodies

**Sarah A. Robinson**[⊕][ʘ], **Matthew I. J. Raybould**[⊕][ʘ], **Constantin Schneider**[⊕], **Wing Ki Wong**[⊕], **Claire Marks**[⊕], **Charlotte M. Deane***

Oxford Protein Informatics Group, Department of Statistics, University of Oxford, United Kingdom

ʘ These authors contributed equally to this work.
* deanestats.ox.ac.uk

**Data Availability Statement:** The properties of the clusters generated in this work are available in the Supporting information files, all model structures are freely available on Zenodo (https://doi.org/10.

## Abstract

Identifying the epitope of an antibody is a key step in understanding its function and its potential as a therapeutic. Sequence-based clonal clustering can identify antibodies with similar epitope complementarity, however, antibodies from markedly different lineages but with similar structures can engage the same epitope. We describe a novel computational method for epitope profiling based on structural modelling and clustering. Using the method, we demonstrate that sequence dissimilar but functionally similar antibodies can be found across the Coronavirus Antibody Database, with high accuracy (92% of antibodies in multiple-occupancy structural clusters bind to consistent domains). Our approach functionally links antibodies with distinct genetic lineages, species origins, and coronavirus specificities. This indicates greater convergence exists in the immune responses to coronaviruses than is suggested by sequence-based approaches. Our results show that applying structural analytics to large class-specific antibody databases will enable high confidence structure-function relationships to be drawn, yielding new opportunities to identify functional convergence hitherto missed by sequence-only analysis.

## Author summary

Antibodies are a key component of the immune system that combat pathogens by binding to a defined region of their molecular surface (known as an 'epitope'). The ability to map which antibodies target the same epitopes is crucial when designing non-competing antibody therapeutics or predicting the influence of pathogen mutation on population immunity. While one can use laboratory experiments to deduce when pairs of antibodies engage the same epitope, such experiments are very expensive and time consuming if used to compare on the order of thousands of antibodies. In this work, we report a new computational algorithm (SPACE) that clusters antibodies that target the same epitope based on their predicted 3D structure, as binding site structure is a property often conserved between binders complementary to the same epitope. Unlike existing antibody epitope profiling tools which assume two antibodies must share a high sequence identity/

5281/zenodo.5569157). The SPACE code is available from the OPIG resources page (http://opig.stats.ox.ac.uk/resources).

**Funding:** This work was supported by an Engineering and Physical Sciences Research Council and Medical Research Council Grant [EP/L016044/1] (MR, SR, WW) and further research funding from F. Hoffmann-La Roche (MR, WW), AstraZeneca plc (CS, MR), GlaxoSmithKline plc (MR) and UCB Pharma Ltd. (MR, SR, WW). The funders had no role in study design, data collection and analysis, decision to publish, or preparation of the manuscript.

**Competing interests:** The authors have declared that no competing interests exist.

similar genetic basis to engage the same region, our orthogonal method can detect broader patterns of convergent evolution across binders to different pathogen strains, and between antibodies with different genetic and even species origins.

## Introduction

The COVID-19 pandemic has generated worldwide efforts to isolate and characterise antibodies able to confer protection against SARS-CoV-2. Hundreds of studies have now released data on diverse antibodies and nanobodies capable of binding at least one coronavirus antigen [1].

Due to the escalating number of individuals infected by SARS-CoV-2, most of the reported coronavirus-binding antibodies to date have been sourced directly from the blood of convalescent human patients. The primary technique used to identify such binders is 'serum baiting', where an extracellular coronavirus antigen is used to pan donated blood serum directly for complementary antibodies [2, 3]. Another increasingly-used method is deep sequencing of the SARS-CoV-2 convalescent B-cell receptor (BCR) repertoire, which can implicate particular expanded antibody lineages as important to adaptive immunity without biasing towards a chosen antigen bait [4–6]. Other discovery methods have included mining surface display libraries, challenging and harvesting transgenic animals, and antibody engineering [1].

As of 11th March 2021, over 2,400 SARS-CoV-2 binding antibodies and nanobodies had been identified, of which just under one-third show neutralisation activity against the virus. The properties (including sequence and, where possible, structure) of these antibodies are documented in the Coronavirus Antibody Database (CoV-AbDab), which tracks patents and the academic literature on a weekly basis [1]. One way to use this collated data is to look for similarities between binders. For instance, when a novel antibody is antigen-baited out of SARS-CoV-2 response serum, or is identified as an expanded clonal lineage post-SARS-CoV-2 infection, one can assess whether it bears resemblance to any other antibody previously reported to bind a coronavirus. This resemblance can then be used to predict functional properties of the newly-isolated antibody, such as its site of engagement with the antigen (the 'epitope').

A common way to cluster antibodies into such functional groupings is 'clonotyping', a form of clonal lineage clustering. This can be performed in several different ways [7]. For example, strict Fv-clonotyping maps both VH and VL antibody chains to their closest immunoglobulin V- and J-gene and subsequently clusters identical gene mappings by their CDRH3 and CDRL3 lengths and sequence identities (using a threshold close to 100% per CDR3 region). This approach typically yields tight and functionally-significant clustering, but is severely limited by its ability to bring together all antibodies able to engage a particular epitope [8–10]. As a result, leniency is often introduced to the clonotyping protocol, by lowering the sequence identity threshold to 80% [11], ignoring J-gene annotations, and/or only considering the heavy chain (VH-only clonotyping) [4].

Convergent lenient VH-only clonotypes have been identified between multiple SARS-CoV-2 infected or convalescent individuals [12–18] and across different studies, for example the overlap between the clonotypes found by Galson *et al*. [4] and Nielsen *et al*. [19]. Several papers have compared BCR sequences from individuals to verified SARS-CoV-2 binders in CoV-Ab-Dab and identified clonal similarities (e.g. [4, 15]).

Despite this, the clonotypes found to be enriched/to bear similarity to CoV-AbDab antibodies post SARS-CoV-2 exposure often differ across studies [20–22]. This may be partly due to the small sample sizes used in individual studies, and the intrinsic biases in individual VJ gene usage; Xiang *et al*. found a larger variation between individuals within a cohort (healthy, or

three different severities of COVID symptoms) than between cohorts [22]. Another contributing factor is lineage clustering itself. Levels of functional convergence may in fact be higher than implied even by lenient clonotyping, as antibodies that derive from different lineages can engage the same epitope. The evidence for this phenomenon has been growing over recent years [9, 10, 23–25]. For example, solved structures of antibody-antigen complexes reveal pairs of antibodies with different genetic origins but sufficiently similar binding site geometry and paratope similarity that they bind to the same antigen with near-identical binding modes [9, 10]. Furthermore, given that individuals' naive repertoires typically share very few clonotypes [11, 26] and yet are often found to respond to similar 'immunodominant' epitopes, it follows that multiple evolutionary routes may lead from low-moderate affinity naive BCRs to high affinity antibodies against the same antigen surface region. This is supported by statistical arguments showing the implausability of a purely "random repertoire" for an efficient immune response [25, 27]. Epitope immunodominance could be rationalised *via* the existence of a more 'public' set of backbone structures in the BCR repertoire and the concept that BCRs with similar topologies and sufficient chemical complementarity engage the same epitope [24, 25].

Structural comparisons offer a way to analyse antibody data over and above clonotype-based approaches. Analysis of epitope regions using solved structures and competition assays of SARS-CoV-2—binding antibodies has already revealed discrete antibody binding sites [28, 29]. However, such assessments are very biased towards receptor binding domain (RBD)-binding, neutralising antibodies, whose therapeutic potential renders them worth the expense and effort of structure determination. The vast majority of datapoints in CoV-AbDab do not have solved structures and must instead be structurally modeled. While homology models are provided alongside each structurally-unsolved CoV-AbDab entry, no studies have yet harnessed this data for functional annotation.

In this analysis, we examine how structurally analysing CoV-AbDab can enhance our functional understanding of coronavirus-binding antibodies. We first analyse all X-ray crystal structures of antibodies/nanobodies bound to SARS-CoV-2 antigens, showing both that structure is conserved more often than clonality across same-epitope binders, and that paratope profiles typically involve multiple regions of the antibody across both chains. This provides direct evidence that relatively sequence dissimilar coronavirus-binding antibodies with high variable domain (Fv) structural similarity are able to exhibit functional commonality. We then model and structurally cluster the thousands of antibody Fv sequences in CoV-AbDab and show that 92% of multiple-occupancy structural clusters bin together antibodies that bind to consistent coronavirus antigens/domains. We also show that, in accordance with our analysis of the SARS-CoV-2 X-ray co-crystal structures, the antibodies within these structural clusters frequently transcend clonal lineages. This not only demonstrates that our computational structural analysis pipeline provides orthogonal information to clonotyping to improve antibody functional profiling, but also that antibody immune responses to SARS-CoV-2 are likely to be even more convergent than currently understood. We chose to apply our method to CoV-AbDab to illustrate the value of structural clustering on an example dataset. Our method could be applied to any high-quality disease-focussed antibody dataset to extract additional information and supplement existing clonotyping analyses.

## Materials and methods

### Database preparation

The version of CoV-AbDab [1] used throughout this analysis was timestamped to the 11th March 2021. The framework and 6 FREAD [30, 31] CDR loop databases, which were used during structural modeling to find suitable homologous templates for each antibody region, were

also timestamped to contain the quality-filtered contents of SAbDab [32] on 11$^{th}$ March 2021. Quality filtering restricts templates to those solved by X-ray crystallography, with a resolution $\leq$ 2.5Å and a B-factor < 80.

## Numbering scheme and region definitions

IMGT numbering [33] is used throughout the manuscript. IMGT CDR region definitions are used to analyse the solved SARS-CoV-2 structures. ABodyBuilder uses North CDR definitions in template selection. The North-defined and IMGT-defined CDR3 region lies between IMGT residue numbers 105 and 117 in both the heavy and light chains, meaning clonotype definitions are consistent regardless of region definition (see Clonotyping).

## Solved co-crystal structure analysis

Sixty solved X-ray co-crystal structures of antibodies and nanobodies bound to SARS-CoV-2 were downloaded from SAbDab. All antibodies were aligned based on the coordinates of the cognate SARS-CoV-2-RBD chain using PyMOL functions. Paratope residues were defined as any antibody residues with a heavy atom with 4.5Å of an antigen heavy atom. We refer to antibodies by name as referenced in CoV-AbDab, using the nomenclature as set out in the literature from which the antibody was sourced. See S1 and S2 Tables for the names and corresponding PDB codes of antibodies.

## Structural modeling and analysis

The 2,063 full variable domain (Fv) sequences in CoV-AbDab were submitted to the ABodyBuilder antibody modelling tool [34] with default Environment Substitution Score cutoffs. In the first instance, ABodyBuilder seeks to model antibody CDR regions entirely by homology; *i.e.* to use the FREAD software [30, 31] to identify a CDR structural 'template' likely to be adopted by each of the submitted antibody's CDR sequences, considering backbone dihedral angle compatibility and loop graftability onto the framework template. If no suitable structural template can be found for a CDR sequence, *ab initio* or hybrid homology/*ab initio* approaches must be used to predict the loop structure, adding uncertainty to model quality. To ensure high model quality, only the 1,500 models for which ABodyBuilder used FREAD to homology model all six CDR loops were carried forward for structural clustering [34].

**Structural clustering algorithm.**   We developed the Structural Profiling of Antibodies to Cluster by Epitope (SPACE) algorithm to structurally cluster the 1,500 models.

These 1,500 Fvs were first split by their combination of six CDR lengths. For each unique CDR length combination, the first antibody in the list was selected as a cluster centre and every subsequent antibody is fed into the following equation:

$$\sqrt{\frac{\sum_X^{(H1-H3,L1-L3)} D^2_{X_{12}} L_X}{\sum_X^{(H1-H3,L1-L3)} L_X}}$$

where the sum over X refers to each of the six CDRs, $L_X$ is the length of North CDRX, and $D_{X_{12}}$ is the $C_\alpha$ RMSD between the template used to model CDRX in Fv 1 and Fv 2 respectively. If this formula equates to $\leq$ 0.75Å, the Fv is clustered with the first cluster centre, otherwise it is held out for the next round of clustering. Once all the Fvs have been considered relative to the first cluster centre, the algorithm progresses in a greedy fashion to select the next unclustered Fv region as the second cluster centre. Greedy clustering was selected due to its

simplicity, good performance, and speed, ensuring scalability across larger datasets. The result is a set of structural cluster centres and associated antibody Fv sequences, where each structural cluster only contains antibodies with six identical CDR lengths. This algorithm is adapted from the final step of Repertoire Structural Profiling [24].

**'Domain-consistent' structural clusters.**   The multiple-occupancy structural clusters were each classified as 'domain-consistent/inconsistent' based on the CoV-AbDab binding metadata of their mapped antibody sequences. For instance, the following examples of structural clusters would each be considered as 'domain-consistent':

1. a structural cluster that only contains antibodies characterised as binding to the same antigen and the same domain (e.g. all shown to bind the RBD of the spike protein).

2. a structural cluster that only contains antibodies characterised as binding to internally-consistent domains (e.g. some antibodies labeled as spike N-Terminal Domain (NTD) binders and others labeled as S1 non-RBD binders; where the S NTD is a subdomain of S1 non-RBD).

3. a structural cluster that contains some antibodies that are characterised as binding to the same domain, and others that bind to the same antigen without domain-level resolution (e.g. 4 antibodies shown to bind the spike RBD, and 2 antibodies shown to bind to the full-length spike protein).

4. a structural cluster that only contains antibodies characterised to bind to the same antigen, but no antibody has domain-level resolution (e.g. 5 antibodies all shown to bind to the full-length spike protein, but none are localised to a particular domain).

The following structural clusters would both be considered 'domain-inconsistent':

1. a structural cluster that contains antibodies shown to bind to different antigens.

2. a structural cluster that contains antibodies that bind to the same antigen, but to inconsistent domains (e.g. 3 antibodies that have been shown to bind the spike RBD and 1 shown to bind the spike S2 domain).

**Clonotyping.**   Clonotyping was performed using an in-house script. Our lenient VH-clonotyping protocol groups Fvs with matching IGHV genes, the same length CDRH3, and $\geq$ 80% CDRH3 sequence identity, while our lenient Fv-clonotyping protocol additionally requires cluster members to have a matching IG[K/L]V gene, the same length CDRL3, and $\geq$ 80% CDRL3 sequence identity. These are lenient clonotyping threshold conditions by community standards [7], as the CDR3 sequence identity threshold is set to its typical lower bound and there is no requirement for cluster members to map to the same IGHJ/IG[K/L]J gene.

## Results

### Sequences and structures in CoV-Ab-Dab

The growth of coronavirus-binding antibody and nanobody data in CoV-AbDab since its public release on 7[th] May 2020 is shown in Fig 1. The antibody plot indicates how the availability of sequence data rose much more rapidly than structural data at the start of the pandemic, stabilising at a level roughly an order of magnitude higher. However, the availability of solved antibody structures increased markedly in October 2020 and has continued to grow at an even faster rate throughout 2021.

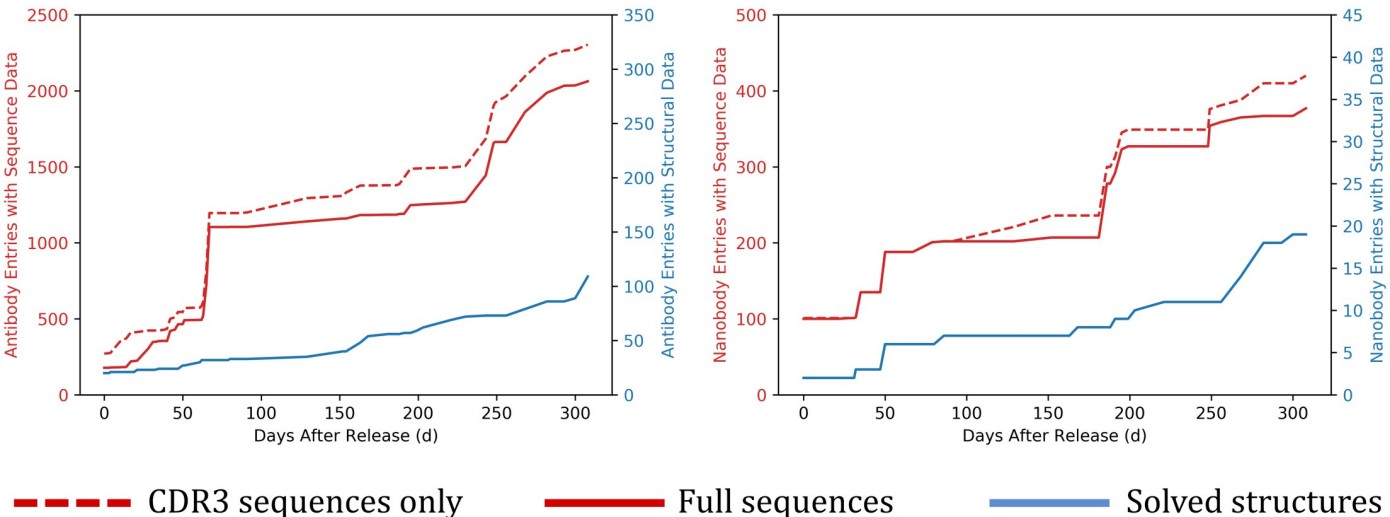

**Fig 1. Comparing the quantity and growth of sequence (red) *vs.* structural (blue) data referenced by CoV-AbDab for antibodies (left-hand plot) and nanobodies (right-hand plot) against any coronavirus antigen.** Structures are classed as solved if evaluated at least once either by X-ray crystallography or cryo-electron microscopy. The x-axis measures the database timestamp after the initial public release of CoV-AbDab on 7th May 2020 (Day 0), up until 11th March 2021 (Day 308). CDR3 sequences are often released ahead of full sequences to protect intellectual property during peer review and/or patent filing.

### Analysis of SARS-CoV-2—Antibody structural complexes

Experimentally-solved structures allow us to analyse the diversity of antibody geometries and paratopes that engage coronavirus antigens [28, 29, 35]. At the time of this study (11st March 2021), CoV-AbDab [1] referenced 2,304 antibodies and 420 nanobodies able to bind to a coronavirus, with 132 having at least one solved structure. 111 of these solved structures are binders to SARS-CoV-2 (Fig 1), of which 91 were solved in complex with the cognate antigen.

A total of 48 antibodies and 12 nanobodies had at least one published solved X-ray crystal structure in complex with SARS-CoV-2 [12, 14, 20, 28, 36–59], all binding to the spike RBD (see S1 and S2 Tables for names and PDB codes), while a further 31 antibodies and nanobodies were solely structurally characterised by cryo-EM [1]. In our analysis we have focused on the 60 crystal structures, in order to determine more precise antibody binding site topologies and paratope profiles.

**Epitope binning.** Inspecting the 48 antibodies solved by X-ray crystallography in complex with the RBD, 46 appear to fall cleanly into the binding regions previously defined by Dejnirattisai *et al.* [28] (only approximate as the original clustering was performed *via* competition assays). The two remaining antibodies spanned the left and right shoulder regions (see S1 Table). Interestingly, most of the 12 nanobodies could also be assigned to these predefined regions (9/12, see S2 Table). For a full analysis of structural epitope overlap between antibodies within epitope groups, see S3 Table.

Structural alignment of the RBD of all complexes reveals that the 22 antibodies that bind to the 'neck' cluster (as termed by Dejnirattisai *et al.* [28]) have high structural conservation (Fig 2A). These antibodies all compete for the ACE-2 receptor binding site (Fig 2B). Dejnirattisai *et al.* identified 13 IGHV3–53/IGHV3–66-derived antibodies engaging this binding site. Even from relatively early in the pandemic, it was clear that a disproportionate number of antibodies reported as able to block ACE-2 binding exploited the IGHV3–53/IGHV3–66 genes [1, 28, 35]; they appeared significantly more often as binders of this region than would be expected by their abundance in healthy antibody repertoires. Banach *et al.* realised that many of the coronavirus-binding antibodies deriving from the IGHV3–53/IGHV3–66 genes possess a

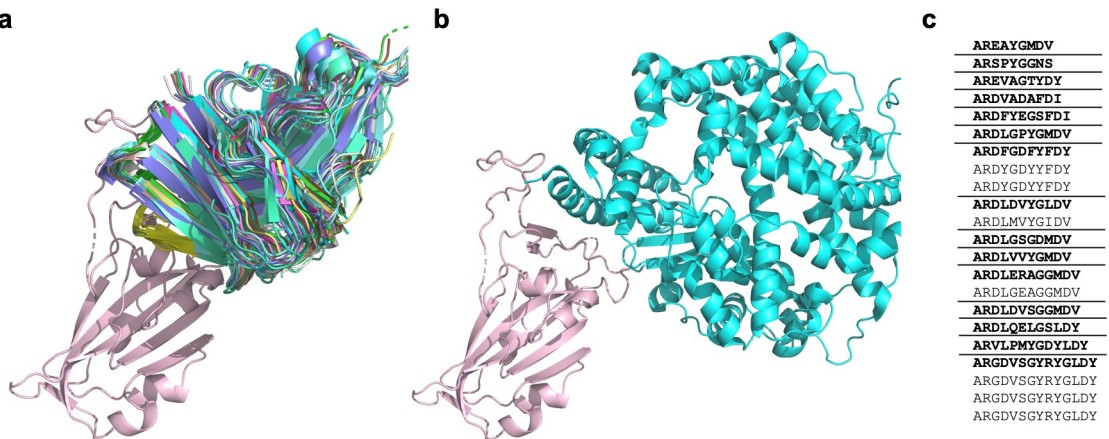

**Fig 2.** a) A cartoon representation of the 22 antibodies in the RBD 'neck' cluster binding to the SARS-CoV-2-RBD (salmon) [PDB code 6XC2] at a similar site to ACE-2. See S1 Table for PDB codes of the 22 antibodies. b) A cartoon representation of ACE-2 (green) binding to SARS-CoV-2 RBD (salmon) [PDB code 6VW1, ACE-2 chain A, SARS-CoV-2-RBD chain E]. C) The CDRH3 sequences represented across the 22 RBD 'neck'-binding antibodies. Lenient VH-clonotypes are separated with solid lines, with the cluster representative highlighted in bold font.

conserved set of structural motifs that enable complementarity to a SARS-CoV-2 RBD epitope [60]. Our updated analysis reiterates the importance of these V gene origins in engaging this highly conserved binding site: 19/22 (86%) of the antibodies align closest to the IGHV3–53 gene, while the remaining 3 align closest to IGHV3–66.

Despite the highly similar V genes, these 22 tightly structurally-clustered antibodies represent 15 different 'lenient VH-only clonotypes' (Fig 2C) (clustered antibodies require the same CDRH3 length, 80% CDRH3 sequence identity, and an identical heavy V gene, see Methods). This corresponds to 18 'lenient Fv clonotypes' when the light chain is also considered (clustered antibodies must also have the same CDRL3 length, 80% CDRL3 sequence identity, and an identical light V gene, see Methods). The analysis of the co-complex structures of these 22 antibodies suggests highly similar functionality, which cannot be wholly identified through clonotyping. Even using lenient clonotype definitions, the antibodies would not be grouped together, so their similar binding mode and functional similarity would be missed.

Dejnirattisai *et al.* [28] also highlighted an antibody binding cluster located away from the ACE-2 binding site, termed the 'left flank'. This binding region contains four antibodies, EY6A, CR3022, S304 and COVA1–16, (see Methods and S1 Table for antibody naming conventions). As shown in Fig 3A, these antibodies appear to bind to slightly different areas. In our updated analysis, we identify one antibody (MW06) and four nanobodies (VHH_U, VHH_V, VHH_W and VHH-72) all able bind to the 'left flank' binding region (Fig 3A). We highlight the antibody pair of EY6A and S304, which are structurally similar and adopt a common binding mode to the same RBD epitope [29], but share only 43% CDRH3 sequence identity (Fig 3B) so could not have been identified as binding to the same site by sequence data alone.

Even across the relatively small number of solved SARS-CoV-2—antibody structures, we can see numerous examples of functionally and structurally similar antibodies that would not be grouped by sequence clustering alone. Grouping coronavirus-binding antibodies into sets that have similar structures therefore represents an orthogonal and promising approach by which to highlight the potential functional commonalities of sequence-dissimilar antibodies.

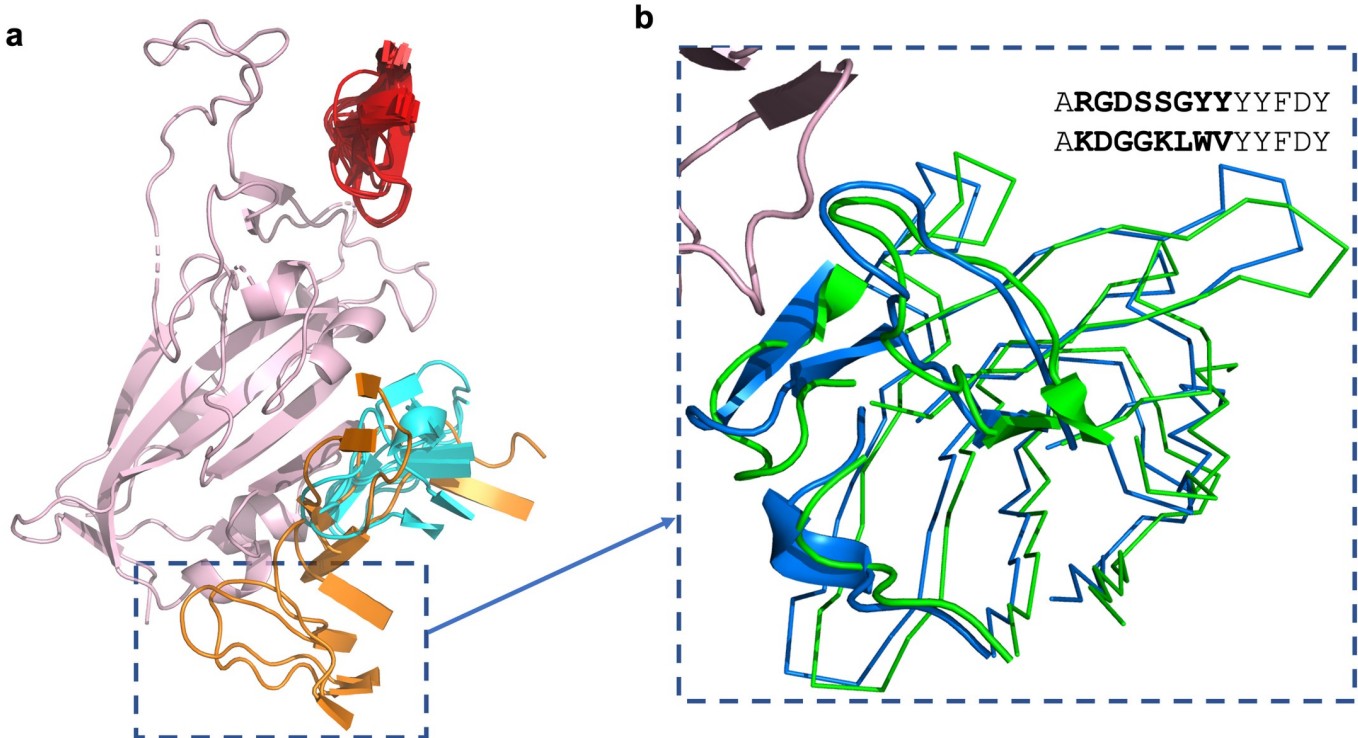

**Fig 3.** a) The CDRH3 loops of nanobodies and antibodies of two binding regions, the 'neck' and 'left flank', binding to SARS-CoV-2 RBD (salmon). The 'neck' cluster includes 22 antibodies (red). The 'left flank' region includes five antibodies (the four antibodies identified by [28] are shown in orange, antibody MW06 is shown in magenta) and four nanobodies (shown in blue). MW06 and the nanobodies were not included in the analysis by [28]. See S1 and S2 Tables for antibody and nanobody PDB codes respectively. b) A ribbbon representation of sequence dissimilar antibodies S304 (blue) [PDB code 7L0N] and EY6A (green) [PDB code 6ZER, chain A] binding to the SARS-CoV-2-RBD (salmon) [PDB code 6ZER, chain B]. The CDRH1, CDRH2 and CDRH3 loops are illustrated in cartoon. The CDRH3 sequences of the two antibodies, S304 and EY6A, are shown, with dissimilar residues indicated in bold.

**Paratope analysis.** We then analysed the binding interfaces across the set of 48 antibodies, to investigate whether structure needs to be conserved across the whole Fv, or whether conservation over particular regions is typically sufficient.

An average of ~67% of paratope residues were found to lie on the heavy (VH) chain while ~33% reside on the light chain (VL). The percentage of paratope residues donated by the most hypervariable region, the CDRH3 loop, varies from just 9% up to 58%. As expected, structures with longer CDRH3 regions tend to exhibit more CDRH3-dominated binding; the paratopes of the nine antibodies with CDRH3 length ≥ 19 on average comprised 41% CDRH3 residues. For the 22 highly structurally-conserved antibodies in the RBD neck epitope region, an average of 23% of the paratope residues originate from CDRH3 (range 8–34%).

These RBD neck-binding antibodies exhibit high levels of paratope residue conservation across the CDRH1 and CDRH2 (Fig 4), which is also seen at the sequence similarity level. This explains the predominance of the IGHV3–53/IGHV3–66 genes, as the residues and topologies pre-encoded by this germline play a key role in neck epitope complementarity. Paratope conservation is considerably less consistent across the CDRH3 region, accounting for the 18 unique CDRH3 sequences seen across the 22 antibodies. For such epitopes, a clonotyping framework (which conditions on high CDRH3 sequence identity being a pre-requisite for same-epitope binding), will clearly fail to capture the functional similarity of the spectrum of cognate antibodies.

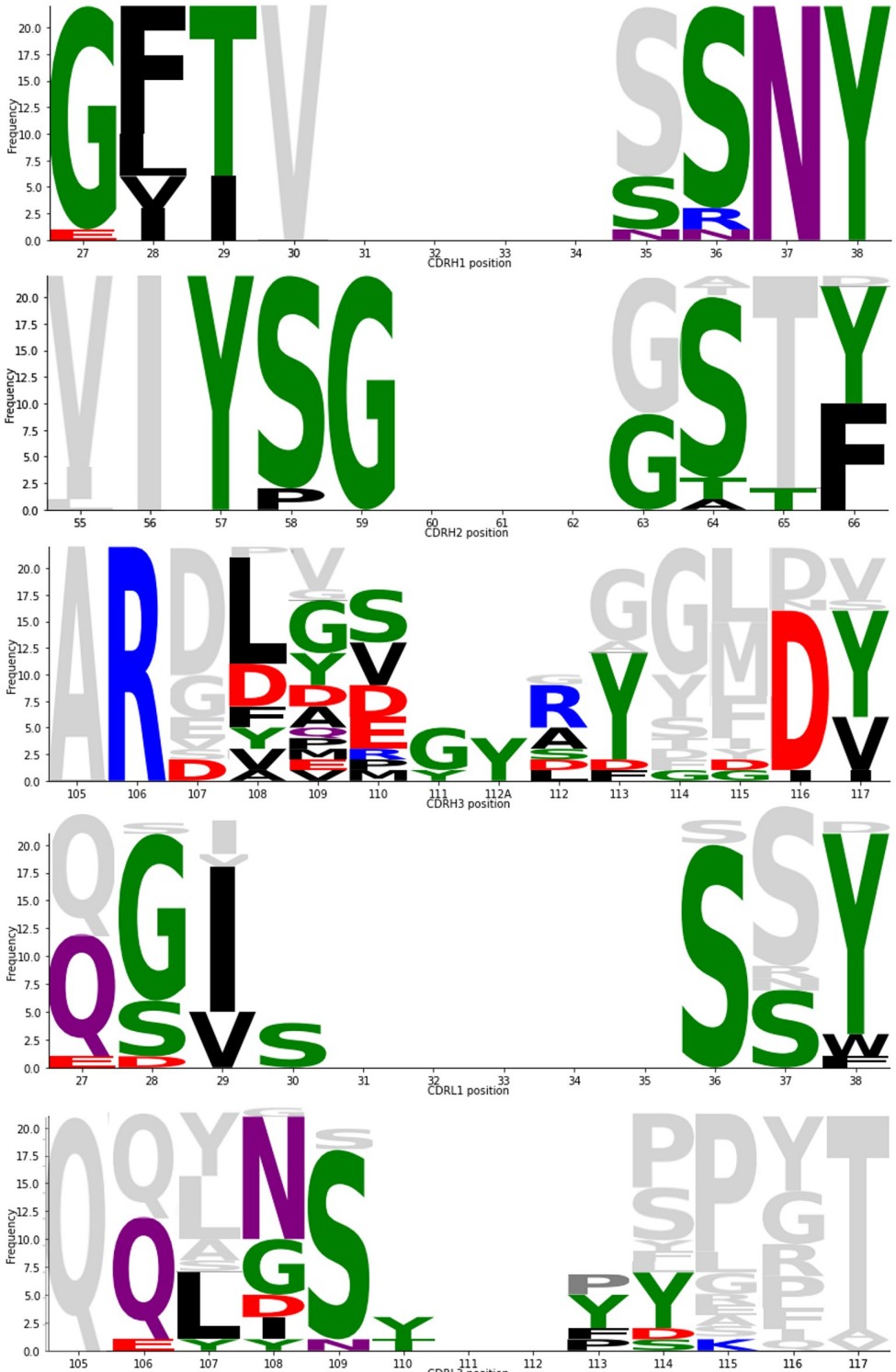

**Fig 4. CDR paratope conservation and divergence across the 22 antibodies in the 'neck' cluster.** Frequency indicates the number of times an amino acid was seen at each IMGT-defined CDR position [33]. Paratope residues are coloured by side chain chemistry (black = hydrophobic, green = polar neutral, red = acidic, blue = basic, purple = amide). Grey indicates amino acids present at the positions but not in the paratope (within 4.5Å of the SARS-CoV-2 antigen). CDRL2 has not been shown, as it was found not to contain paratope residues. Produced using Logomaker [61].

For those antibody paratopes that use CDRH3 sparsely, the paratope is mostly drawn from the CDRH1, CDRH2, CDRL1 and CDRL3 residues, with occasional amino acids provided by the framework regions (FWRs). In particular, a serine residue at position 83 and/or glycine at position 84 of the light chain (part of the formal FWRL3, IMGT numbering) are disproportionately involved in binding. These residues were found in 18/48 (38%) of crystallised-antibody paratopes, of which 15 are from the 'neck' structural cluster. Using Arpeggio [62] to identify the type of binding interactions these FWRL3 residues were involved in, 9 paratopes contained hydrogen bonds between L83 and the RBD, to residue 498 in all cases apart from one (for full Arpeggio analysis of interactions between antibodies and the RBD see S3 Table).

Overall, we conclude that structural conservation is important across the entire Fv, to ensure that the paratope residues, which in most coronavirus-binding antibodies are spread evenly throughout the CDRs, are held in equivalent topological positions.

We now perform a comprehensive structure-based analysis on CoV-AbDab, where 95% of SARS-CoV-2 binding antibodies have no solved structure and 27% completely lack binding-domain annotation.

## Structural convergence across CoV-AbDab antibodies

As of 11th March, just ~5% (113/2,304) of the antibodies in CoV-AbDab had at least one solved X-ray or cryo-EM structure, while ~90% (2,063/2,304) of the antibodies had full Fv amino acid sequences (Fig 1). We used high-throughput homology modelling approaches to approximate and analyse the geometries of this much broader set of neutralising and non-neutralising antibodies able to bind to multiple coronaviruses, antigens, and domains.

We used ABodyBuilder [34] to homology model the 2,063 antibody entries in CoV-AbDab with full Fv sequences. This resulted in a total of 1,500 models in which every loop was entirely FREAD-modellable without any need for *ab initio* loop modelling or backbone adjustment (see Methods); we focus on this subset of models as we have the highest confidence in their accuracy [23, 24, 34]. This represents 72.7% modellability across the set of Fv sequences, a remarkably high percentage relative to recent studies on both healthy and disease-related natural antibody datasets [23, 24]. Typically, only ~40% of randomly-sampled human CDRH3s can be homology modelled by FREAD. The increase in modellability is likely to be related to the scientific effort that has gone into solving a large number of SARS-CoV-2-binding antibody structures within the first year of the pandemic. It also hints at a high degree of underlying structural convergence across the reported coronavirus-binding antibodies in CoV-AbDab.

After modelling, we performed structural clustering. Briefly, Fvs with 6 modellable CDRs are first clustered according to their combination of six CDR lengths, and are then further structurally grouped by a greedy clustering algorithm that considers the pairwise structural root-mean square deviation (RMSD) between the selected FREAD template for each CDR region (see Methods for a full description). The result is a set of predicted 'structural clusters', each adopted by at least one Fv sequence. We term this algorithm Structural Profiling of Antibodies to Cluster by Epitope (SPACE); see Methods for a full description.

The 1,500 homology modelled Fv regions fell into 1,159 structural clusters, of which 200 were adopted by more than one Fv sequence ('multiple-occupancy' structural clusters). In total, 541/1500 (36.1%) of the antibodies belonged to a multiple-occupancy structural cluster. For a full breakdown of each multiple-occupancy structural cluster, labelled SC0-SC199, see S1 File.

When applied to an entire antibody repertoire, the number of antibodies with similar structures but different functionalities is likely to be significant [24]. However, on 'cleaner' datasets

such as CoV-AbDab, where every antibody has been shown to bind a coronavirus antigen (and often a particular domain) *in vitro*, antibodies with similar Fv structures should have a high chance of binding to the same surface region and therefore being complementary to the same epitope, as demonstrated in our analysis of solved structures.

Nonetheless, inaccuracy in CDR structure prediction and/or the lack of consideration for paratope residues can lead to misleading clusters of antibodies that have markedly different functionalities. Therefore, to assess (in the context of this large disease-specific antibody database) how predictive belonging to the same structural cluster is of engaging the same epitope, we estimated a pseudo-'true positive'/'false positive' ratio based on the consistency of the CoV-AbDab epitope metadata across antibodies grouped into the same structural cluster. 'Domain-consistent' structural clusters were classed as those only containing antibodies reported as binding to internally-consistent antigen domains (see Methods).

A total of 184/200 (92%) of our multiple-occupancy structural clusters were domain-consistent, indicating that structurally clustering with another member of CoV-AbDab is likely to be highly predictive of function.

Amongst the 16 structural clusters considered to be 'false positives', the domain-inconsistent antibody within four structural clusters (SC2, SC84, SC96, SC136) bore significant similarities to at least one other antibody in the cluster, suggesting that some experimentally-deduced epitope labels may be inaccurate. The original papers detailing the binding of the antibodies support this possibility. For example, we classify SC2 as domain-inconsistent, as it contains 8 antibodies that bind to spike outside the RBD [63, 64], and one (COVA2–32 [65]) labelled as binding to the RBD. However, COVA2–32 only marginally met the area under curve (AUC) threshold to be classified as an RBD binder (see Fig 4 of Brouwer *et al.* [65]).

The remaining twelve 'false positive' structural clusters are likely to result from a combination of inaccurate structural modelling and/or the fact that bearing the same binding site structure does not guarantee functional commonality. We focus the remainder of our analysis on the 92% of domain-consistent structural clusters.

**Epitope binning.**   Some members of a structural cluster can have a lower resolution of functional characterisation than others. In these cases, functional properties of the less well-characterised antibodies can be inferred from other antibodies predicted to adopt the same structure.

Thirty-one antibodies experimentally shown only to bind to the whole spike protein, or to bind the spike protein but not the RBD, can be localised to a more precise epitope using our structural clusters. For example, three of the antibodies assigned to SC11 that were shown to bind the full-length spike protein, but not a soluble RBD protein [63], can be inferred to bind to the S2 domain in the same way as cluster members DH1147 and DH1149 [64]. Similarly, CC12.24, previously shown only to bind to the whole SARS-CoV-2 spike protein [41], can be localised to the same binding site in the RBD as C139 [12] and COVOX-45 [28] (SC57).

A further 62 antibodies fall into 19 structural clusters for which no antibody has been resolved as binding to a particular domain. For some structures, the selected FREAD templates could offer an indication of epitope specificity: the PDB structure '6NB8' [66] is used to model all three light chain CDRs in at least one antibody assigned to SC6, SC15, and SC71, while '7BEN' [28] is used to model the CDRH3 in all four antibodies assigned to SC16. The antibody in both of these PDB structures engages the SARS-CoV-2 RBD, which could imply that the cluster groups RBD-complementary antibodies. Even ignoring these indications, just 19 binding characterisation experiments could lend functional annotations to over three-times the number of CoV-AbDab entries (62).

**Evolutionarily-conserved epitope topologies across coronaviruses.**   Evolutionarily-conserved epitope topologies are implied by our structural clusters that contain antibodies able to

bind multiple coronaviruses. As an extreme example, SC0 pools together a set of 13 IGHV1–69-derived antibodies, of which at least one which has been shown to engage each of SARS-CoV-1, SARS-CoV-2, HKU1, 229E, NL63, OC43, and MERS-CoV. More broadly, 69/184 (37.5%) of our structural clusters contain an antibody shown on the current levels of data to have cross-coronavirus binding potential. This number may be an underestimate, as several antibodies have only been tested against a single coronavirus strain *in vitro*.

These epitopes could represent sites of particular vulnerability across coronaviruses; antigen regions whose structure must be preserved for viral function. They therefore reflect the most promising regions against which to design pan-coronavirus neutralising antibodies, and could be exploited in epitope-focused vaccine design strategies [67] to achieve a more broadly neutralising response.

The epitope and corresponding paratope residues within these binding sites will differ between lineages, meaning that sequence-based clustering approaches would struggle to spot their functional commonality. Structural clustering can see beyond paratope profiles to capture broader epitope topology conservation *via* the geometries of their cognate antibodies.

**Epitopes targetable by multiple species.**   A recent study has shown that mouse and human antibodies use a different distribution of CDRH3 structures, which also varies by B-cell maturation stage [23]. This can be rationalised by their species-specific gene loci having different predetermined structural biases and the fact that negative selection occurs against different self-epitopes. Nevertheless, assuming that each CDR loop can only adopt a finite set of geometries imposed by the loop closure criterion, and that many of the same antigens would be considered pathogenic to both species, there ought to be some structural overlap between human and mouse antibodies and therefore the potential for some epitopes to be targetable by both species. These would be extremely hard to identify by sequence alone, as human and murine gene loci are highly sequence dissimilar.

We identified SC62, which groups human antibody Ab_511E7 [68] alongside the two murine antibodies DK4 & DK7 (patent CN111978395A), all of which have been shown to bind the SARS-CoV-2 RBD (albeit only weakly in the case of Ab_511E7). Even more remarkably, SC174 pools antibodies from different species confirmed to bind to different coronaviruses; in SC174, a human antibody (C131 [12]) that binds the SARS-CoV-2 RBD is grouped with a murine antibody (F26G18 [69]) shown to bind the full-length SARS-CoV-1 spike protein. Should F26G18 be confirmed to engage the RBD, and the SC62 antibodies compete for the same epitope, this would show the ability of structural clustering to identify cross-coronavirus epitopes targetable by multiple organisms.

**Structural clusters frequently span multiple clonal lineages.**   We analysed each of the 184 domain-consistent structural clusters to determine how often the antibodies clustered together belonged to multiple lenient Fv clonotypes. A total of 88 (47.8%) contained at least one pair of antibodies from different lenient Fv clonotypes and 73 (39.7%) of the structural clusters contain at least two lenient VH-only clonotypes. It is clear that antibodies with both heavy and light chains of differing clonality can frequently co-exist within our structural clusters.

Many structural clusters contain at least one pair of antibodies from the same clonotype; this is unsurprising since the 'near-identical sequence, similar function' assumption underpinning clonotyping experiments is often correct. However, the high frequency with which we group antibodies spanning several clonotypes into the same structural cluster recapitulates the findings of other papers [8–10], and our earlier analysis on solved structures, that clonotyping cannot group together all antibodies capable of same-epitope engagement.

In most cases where multiple clonotypes are found in the same structural cluster, it is due to significant differences in the CDRH3 sequence. However, some clusters such as SC134 (which pools COV2–2490 (60) with H712061+K711727 (61)), contain many differences across

the entirety of the sequence (26 differences across VH, 27 across VL) and align closest to different heavy V (IGHV3–7 vs. IGHV3–30) and light V (IGKV1–5 vs. IGKV1D-16) genes.

**'Public' response antibodies.** 'Public' antibodies, those that are raised independently across multiple individuals against an immunodominant epitope, are of high interest to several fields of research, from vaccinology to drug discovery [70–72].

Several studies have already identified public SARS-CoV-2 response antibodies based on convergence towards particular clonotype lineages [12–18], but none have yet considered the fact that antibodies from different lineages can exert similar functions. As demonstrated above, structural clustering enables us to group together clonally-distinct antibodies with a high chance of engaging the same epitope. We therefore examined our structural clusters to reveal functionally similar groups of antibodies from different genetic lineages that have been independently isolated across several different studies (*i.e.* "public structures").

A striking example of a public structure that spans multiple clonotypes is SC3 (Fig 5). SC3 contains nine antibodies from five independent sources, spanning five lenient Fv-clonotypes. All antibodies align closest to the IGHV1–58/IGKV2–30 gene transcripts but have sequence-diverse CDRH3s all containing a common intra-loop disulfide bridge.

The SC21 and SC24 clusters also map to these genes and contain a disulfide bridge; antibodies assigned to SC21 and SC3 have identical CDR lengths but are predicted to have different CDR structures, while SC24 is necessarily classed as a separate structural cluster to SC3 as its CDRL3 loop is of a different length (10 residues rather than 9).

When aligned to the solved COVOX-253:RBD co-crystal structure, SC3, SC21, and SC24 appear to all be topologically complimentary to the same RBD epitope (Fig 5). The length-10, more protruding CDRL3 loop of SC24 is accommodated by the small G485 residue on the RBD, while the CDRH3s across all three structural clusters protrude to a similar extent towards residues F456-N460 on the RBD. The antibodies mapped to SC21 and SC24 (from an additional three independent sources) comprise an additional three lenient Fv-clonotypes, making a total of eight lenient Fv-clonotypes with potential same-epitope complementarity.

This set of similar structures has been observed across eight independent studies indicating that the corresponding epitope is immunodominant. Moreover, none of the antibodies directly engage the carboxylate group of residue E484 (5Å for length-10 CDRL3s or 11Å for length-9 CDRL3s, with too acute an angle for hydrogen bonding) nor the amide group of N501 ($\geq$12Å for all antibody topologies). This should make them of particular interest as clones that might neutralise both wildtype SARS-CoV-2 and the more recent E484K/N501Y-containing variants of concern.

Soon after we identified this broad structural cluster, a preprint was released by Schmitz *et al.* [73] showing that many IGHV1–58-encoded SARS-CoV-2 binding antibodies have highly similar residues at equivalent paratope positions (defined by the S2-E12 crystal structure [74]). On this occasion, the other CDRH3 sequence-diverse antibodies hypothesised to have similar function were shortlisted through inspection of their sequences; they all derive from IGHV1–58 germline and bear the -C(X)$_4$C- motif within their CDRH3 loop (X$_4$ representing four non-cysteine residues). We have shown that structural modelling and clustering supports the theory that these antibodies are functionally similar, and offers a systematic route to the identification of other sequence-diverse clusters of functionally-common antibodies that do not bear such clearly conserved motifs.

## Discussion

Here, we have analysed the solved X-ray crystal structures of antibodies and nanobodies bound to SARS-CoV-2 from the perspective of their structural and paratope conservation, and

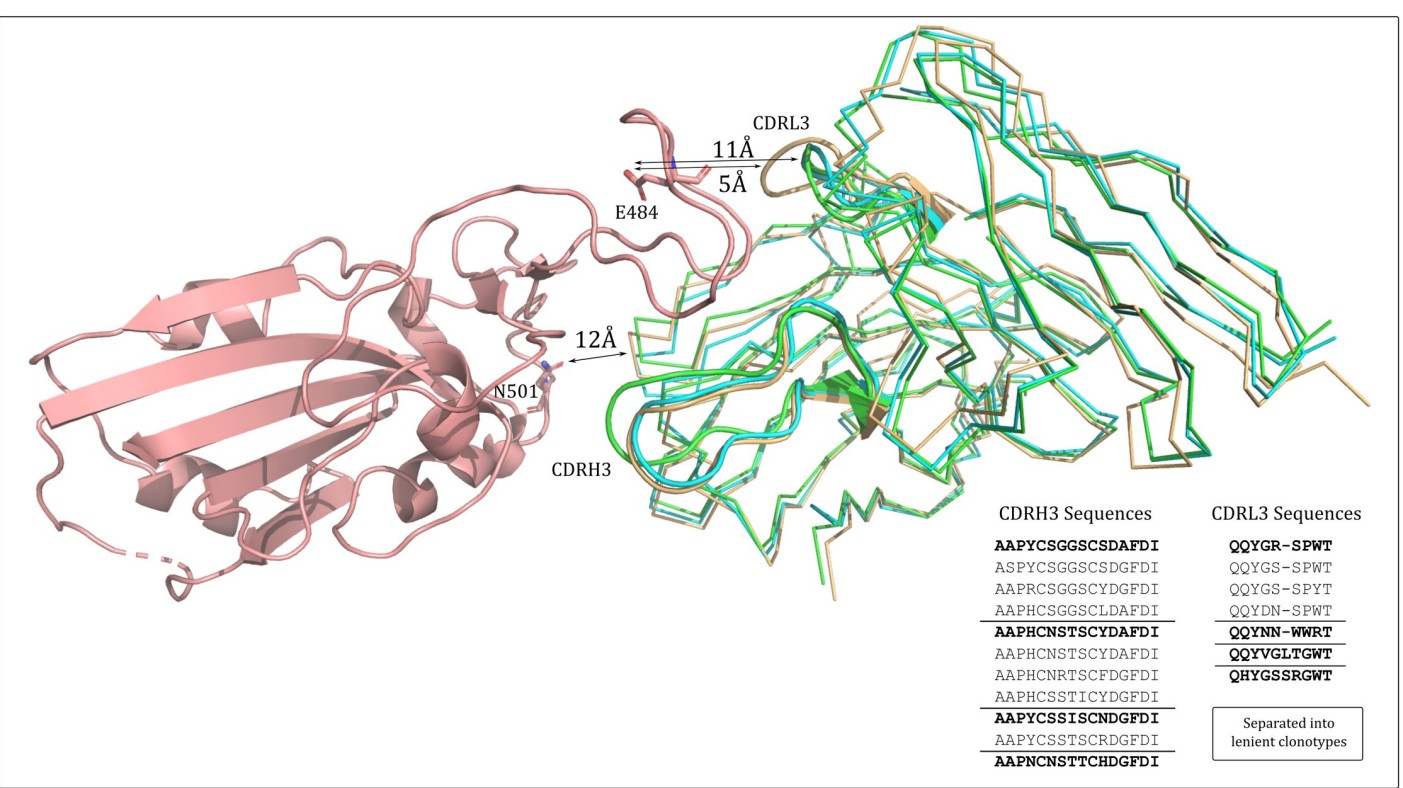

**Fig 5. SARS-CoV-2 RBD-binding antibodies with similar predicted structure that span multiple clonotypes.** A representative of structural cluster 3 (SC3, cyan), structural cluster 21 (SC21, orange), and structural cluster 24 (SC24, green) are aligned in the context of the COVOX-253:RBD co-crystal complex (RBD in salmon). The CDRH3 and CDRL3 structures are highlighted in cartoon representation. All 100% sequence non-redundant CDRH3 and CDRL3 sequences across the three structures are listed, grouped by lenient VH- or VL-clonotype, with the cluster representative in bold font. The alignment shows the various CDRH3 and CDRL3 structures are likely to be topologically compatible with this RBD epitope. Residues E484 and N501, commonly mutated in SARS-CoV-2 variants of concern, are highlighted as sticks and coloured by default atom type. Closest heavy-atom distances between the functional group (carboyxlate/amide heteroatom) and the different structural classes of antibody are shown.

performed the first analysis of thousands of structural models of coronavirus-binding antibodies reported in over 100 independent literature sources.

We have updated the previously-reported sets of antibodies shown to bind to specific regions of the RBD [28] and demonstrated that the antibodies within these clusters are often structurally similar but sequence dissimilar. For example, a cluster of 22 antibodies from 15 different lenient VH clonotypes all approach the same SARS-CoV-2 RBD 'neck' epitope with a closely-related binding mode. The paratopes of these antibodies are highly conserved across the CDRH1 and CDRH2 regions (accounting for the strong bias towards IGHV3–53/IGHV3–66 gene origins) while the CDRH3 sequence can diverge substantially in sequence identity. The same phenomenon was observed for two antibodies solved engaging the RBD 'left flank' with a near-identical binding mode but just 43% CDRH3 sequence identity. These scenarios represent a problem for the functional interpretation of sequence-based clustering approaches such as clonotyping; antibodies that are functionally similar would be binned into different clusters. Inspired by the structural similarity of antibodies that bind to the same epitope, we predicted and clustered the structures of the broad set of antibodies documented in CoV-Ab-Dab [1] using our SPACE algorithm.

We found that this structural clustering is likely to achieve a very high accuracy of epitope binning. Up to 92% of multiple-occupancy structural clusters grouped antibodies reported to

bind to consistent domains, based on the current levels of metadata in CoV-AbDab. This suggests that we can use this method to predict the epitopes of many as-yet uncharacterised coronavirus binders, as well as prospectively to predict the epitopes of newly-isolated SARS-CoV-2 binding antibodies.

These structural clusters also offer a unique perspective on the data that can not be identified through standard, sequence-based clonotyping. First, they can functionally-associate antibodies that derive from highly distinct clonal lineages; around 40% of our structural clusters contain at least two antibodies from different lenient VH clonotypes. They also functionally connect disease-response antibodies that originate from different species, of interest in the study of functional crossovers between immune repertoires that exploit different gene loci. Moreover, they can reveal which epitope topologies are likely to be conserved across coronavirus strains, helping to co-ordinate efforts to design prophylactics towards more fruitful sites for pan-coronavirus neutralisation.

Such powerful structure-function relationships are likely only possible due to the collation of clean, high confidence binding data. Our work demonstrates the clear value of building large class-specific databases of antibody-binders against extracellular disease-associated antigens. We hope to create similar datasets for other pathogen families and suspect others will do the same, enabling the validation of computational structural profiling in multiple different disease contexts.

In addition to offering functional annotations to as-yet uncharacterised antibodies, these databases could also be used to identify important gaps in structural space. This would ensure that time-consuming experimental structure determination efforts are targeted towards sets of antibodies that yield maximal functional insight. For structural clusters with limited existing epitope knowledge, a central antibody could be selected for structural evaluation with the cognate antigen, enabling the functional annotation of many other antibodies in the same structural cluster. Similarly, structurally modelling the entire database reveals which antibodies cannot currently be accurately modeled and should therefore be prioritised for experimental structure characterisation.

As more coronavirus-binding antibody structures continue to be released to the PDB, the coverage and expected accuracy of the CoV-AbDab homology models will increase accordingly, likely further improving the accuracy of our epitope binning over time. However, solved antibody-pandemic virus structures are not a prerequisite of meaningful epitope binning *via* predicted structure. Forty-eight of our domain-consistent structural clusters currently connect antibodies that bind outside of the RBD, despite the fact no high-quality X-ray structures of antibodies binding outside the SARS-CoV-2 RBD had been solved at the time of this analysis ($\leq 2.5$Å resolution, a requirement for use as an ABodyBuilder template). This indicates that structural clustering is able to draw functional connections between antibodies isolated at the start of a pandemic, even if the number of solved antibody-pandemic virus structures is very low.

Additionally, given recent developments in single-domain protein modelling techniques [75], we may soon see significant improvements in the speed and accuracy of antibody structure prediction, and by extension antibody structure-based epitope profiling. Many alternative clustering methods exist beyond the template-based approach reported here, and these may be more appropriate to use when clustering models generated by future high-throughput structure modelling algorithms.

An open question remains as to how strictly an antibody's structure needs to be conserved to engage the same epitope. This is likely to be highly epitope dependent. For example, for epitopes naturally suited to VH-dominated engagement, less selection pressure would act upon light chain structure and vice versa, while some epitope topologies and environments (e.g.

extent of glycosylation) may also exert different levels of pressure on complementary antibody geometries.

Overall, our results show that structural information *via* computational modeling enhances the picture of disease-characteristic convergence across SARS-CoV-2 response antibodies. It is clear that while clonotyping COVID-19 antibody repertoires can offer an indication for which epitopes are public in the response, they risk understating the true levels of same-epitope reactivity across individuals. Accurately capturing the functions of the antibodies raised during the immune response is critical when evaluating vaccine efficacy, both against the injected viral strain and against new variants of concern that may arise and alter the immunogenicity of certain epitopes. Structure prediction and clustering has a crucial role to play alongside clonotyping to yield the maximum functional inference from the vast amount of disease-specific antibody data available.

## Supporting information

**S1 File. Structural clusters.** Multiple occupancy structural clusters identified using SPACE. (XLSX)

**S1 Table. SARS-CoV-2 binding antibodies.** Antibodies with a solved crystal structures in complex with a SARS-CoV-2 antigen. Antibodies were assigned to their nearest epitope cluster, as defined by Dejnirattisai *et al.* [28]. Antibodies whose epitope cluster is labelled with a star (*) were not included in the previous analysis by Dejnirattisai *et al.* [28]. The new cluster label 'Shoulder' indicates that antibody's epitope lies between the left and right shoulder clusters.
(PDF)

**S2 Table. SARS-CoV-2 binding nanobodies.** Nanobodies with a solved crystal structures in complex with a SARS-CoV-2 antigen. Nanobodies were assigned to their nearest epitope cluster, as defined by Dejnirattisai *et al.* [28]. Nanobodies whose epitope cluster is labelled with a star (*) were not included in the previous analysis by Dejnirattisai *et al.* [28]. The new cluster label 'Shoulder' indicates that antibody's epitope lies between the left and right shoulder clusters.
(PDF)

**S3 Table. SARS-CoV-2 binding antibodies epitope interactions.** Analysis of structural epitope overlap between epitope group members. Epitope residues were defined as those within 4.5A of the antibody in the crystal structure. Analysis was conducted using Arpeggio [62].
(PDF)

**S4 Table. SARS-CoV-2 binding antibodies paratope interactions.** Analysis of the paratope-epitope interactions between antibody and SARS-CoV-2 RBD in the cocrystal structures, determined using Arpeggio [62].
(PDF)

## Author Contributions

**Conceptualization:** Sarah A. Robinson, Matthew I. J. Raybould, Constantin Schneider, Wing Ki Wong, Claire Marks, Charlotte M. Deane.

**Data curation:** Sarah A. Robinson, Matthew I. J. Raybould, Constantin Schneider, Wing Ki Wong.

**Methodology:** Sarah A. Robinson, Matthew I. J. Raybould.

**Software:** Sarah A. Robinson, Matthew I. J. Raybould.

**Supervision:** Claire Marks, Charlotte M. Deane.

**Writing – original draft:** Sarah A. Robinson, Matthew I. J. Raybould.

**Writing – review & editing:** Charlotte M. Deane.

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
