## [Decision Letter · Decision Letter 0]

27 Sep 2021

Dear Prof. Deane

Thank you very much for submitting your manuscript "Epitope profiling of coronavirus-binding antibodies using computational structural modelling" for consideration at PLOS Computational Biology. As with all papers reviewed by the journal, your manuscript was reviewed by members of the editorial board and by several independent reviewers. The reviewers appreciated the attention to an important topic. Based on the reviews, we are likely to accept this manuscript for publication, providing that you modify the manuscript according to the review recommendations.

Sincerely,

Franca Fraternali

Guest Editor

PLOS Computational Biology

Rob De Boer

Deputy Editor

PLOS Computational Biology

[LINK]

Dear Prof.Deane,

your manuscript has been evaluated by three referees that have expressed an overall appreciation of your work, but have also raised some concerns and points to be addressed.

Please reply to these comments carefully and clarify the raised issues.

Reviewer's Responses to Questions

**Comments to the Authors:**

Reviewer #1: This paper by Robinson and colleagues explores structural-based clonal clustering. They show that taking into account structural properties reveals insights not apparent at the sequence level, which is both consistent with increasing bodies of work in the area, but also intriguing. While they focus mostly on COVID targeting antibodies, this should more be seen as a powerful dataset to provide insights that are much more broadly applicable.

The work is well written and will be of broad interest. My two main comments relate to validation and replication.

Validation - The authors illustrate the potential power of structure based clustering using spike rbd antibodies, but to reflect the broad title it should be validated across other antibody sets.

Replication - I would encourage the authors to make their scripts available to enable replication of the work (e.g. reference a github repository rather than just saying that internal scripts were used). This will also facilitate others to apply similar approaches to other datasets.

Reviewer #2: Robinson et al. describe a novel computational method for epitope profiling applied to SARS-CoV2 antibodies using structural clustering coupled with homology modelling, demonstrating a strong structural-functional relationship (that sequence-based clustering would fail to pick up). The work tackles a very relevant and topical problem, particularly in a pandemic context, and deserves publication.

A few points to be addressed:

* Are there any impacts on recent AlphaFold2/RosettaFold developments for this work that are worth commenting (or pursuing further analyses)? I understand this study predates the release of these two methods, but it might be a good opportunity to comment on implications at least.

* Lines 280-284. The statement around interaction analysis is limited to hydrogen bonds and a bit vague ("almost always"). Perhaps it would be interesting to quantify conservation of different interaction types, rather than just Hydrogen Bonds (could also provide a figure highlighting them).

* It would be interesting for the community to make the homology models generated available.

* Was there any sort of energy minimisation performed on the models?

Please clarify.

* Was there any sort of quality control performed on the models?

Please clarify.

* Have the authors tested different clustering approaches? (or what's the rationale behind choosing the one used) How would that affect their conclusions?

Minor Points:

* Typo in the abstract: confidence -> confidence

* Figure 3 (specially panel b) and 4 look low-res

* Line 209 - There is no panel c in Figure 3. I am guessing it is a typo (Figure 2C instead?)

* Line 372 - AUC (Area under curve? - please define abbreviation)

Reviewer #3: There are a few points that I think need to be addressed:

1. Arguably the IGHV3-53/IGHV3-66 example (line 205) is really an argument against the excessive stringency of the “identical heavy V gene” requirement, given that this particular pair of germline genes have near-identical sequences (the IMGT sequences for the *01 alleles of these V genes differ by only a single residue out of 97). The later example (line 489) of IGHV3-7 vs. IGHV3-30 (12 residue differences) is potentially more compelling.

2. There might be several ways to define “same” epitope (line 6 and others). It would be good to have some insight into how loose the working definition is here, e.g. by quoting the overlap of the structural epitopes derived from the structures of bound antibodies considered to bind to the same epitope.

3. There is no direct discussion of epitope constraints — some topologies, or the presence of glycans, may constrain the viable positions and orientations of an antibody more than others, making structural (and sequence) similarity more likely. Presumably some evolutionarily-conserved epitope topologies may be associated with tighter constraints than others, and that is likely to have an impact on the associated antibody structural clusters. This consideration appears relevant to the discussion about conservation and vulnerability (from line 411), and should be addressed.

Two minor points:

1. The abbreviation RBD is defined after its first usage in line 119.

2. There is a typo: confidenfce

**Have the authors made all data and (if applicable) computational code underlying the findings in their manuscript fully available?**

Reviewer #1: **No: **Code is not available

Reviewer #2: **No: **I would encourage the authors to make their scripts available to enable replication of the work.

Reviewer #3: Yes

PLOS authors have the option to publish the peer review history of their article (what does this mean?). If published, this will include your full peer review and any attached files.

Reviewer #1: No

Reviewer #2: No

Reviewer #3: No

Figure Files:

Data Requirements:

Reproducibility:

References:

---

## [Decision Letter · Decision Letter 1]

22 Nov 2021

Dear Deane,

We are pleased to inform you that your manuscript 'Epitope profiling using computational structural modelling demonstrated on coronavirus-binding antibodies' has been provisionally accepted for publication in PLOS Computational Biology.

Best regards,

Franca Fraternali

Guest Editor

PLOS Computational Biology

Rob De Boer

Deputy Editor

PLOS Computational Biology

Dear Prof. Deane,

we are pleased to inform you that the reviewers have ben satisfied by your rebuttal and that your paper is now accepted for publication.

With my best regards

Franca Fraternali

Reviewer's Responses to Questions

**Comments to the Authors:**

Reviewer #1: The authors have done a great job addressing all my concerns, and I think the paper and work read well and are very interesting

Reviewer #3: No further modifications required.

**Have the authors made all data and (if applicable) computational code underlying the findings in their manuscript fully available?**

Reviewer #1: Yes

Reviewer #3: Yes

PLOS authors have the option to publish the peer review history of their article (what does this mean?). If published, this will include your full peer review and any attached files.

Reviewer #1: No

Reviewer #3: No

---

## [Editor Report · Acceptance letter]

7 Dec 2021

PCOMPBIOL-D-21-01018R1 

Epitope profiling using computational structural modelling demonstrated on coronavirus-binding antibodies

Dear Dr Deane,

I am pleased to inform you that your manuscript has been formally accepted for publication in PLOS Computational Biology. Your manuscript is now with our production department and you will be notified of the publication date in due course.

With kind regards,

Zsanett Szabo
